

# Unravelling the microphysics of polar mesospheric cloud formation

Denis Duft[1], Mario Nachbar[1], Thomas Leisner[1,2]

[1] Institute of Meteorology and Climate Research, Karlsruhe Institute of Technology, Karlsruhe, 76021, Germany
[2] Institute of Environmental Physics, University of Heidelberg, Heidelberg, 69120, Germany

*Correspondence to*: Denis Duft (denis.duft@kit.edu)

**Abstract.** Polar mesospheric clouds are the highest water ice clouds occurring in the terrestrial atmosphere. They form in the polar summer mesopause, the coldest region in the atmosphere. It has long been assumed that these clouds form by heterogeneous nucleation on meteoric smoke particles which are the remnants of material ablated from meteoroids in the upper atmosphere. However, until now little was known about the properties of these nm-size particles and application of the

classical theory for heterogeneous ice nucleation was impacted by large uncertainties. In this work, we performed laboratory measurements on the heterogeneous ice formation process at mesopause conditions on small ($r$=1 to 3 nm) iron-silicate nano-particles serving as meteoric smoke analogues. We observe, that ice growth on these particles sets in for saturation ratios with respect to hexagonal ice below $S_h$=50, a value that is commonly exceeded during NLC season, affirming meteoric smoke particles as likely nuclei for heterogeneous ice formation in mesospheric clouds. We find that ice formation on iron-

silicate nano-particles occurs by condensation of amorphous solid water rather than by nucleation of crystalline ice and that amorphous solid water has to be considered as a relevant ice polymorph in polar mesospheric cloud formation. We present a simple ice activation model that takes into account the water affinity of iron-silicates of various compositions. For particles with radius larger than r=1 nm no significant effect of the nano-particle charge on the activation threshold was found.

## 1 Introduction

Polar mesospheric clouds (PMC) are water ice clouds occurring in the terrestrial atmosphere at an altitude of about 83 km (e.g. Rapp and Thomas, 2006; Thomas, 1991; Thomas and Olivero, 2001). The clouds form during the polar summer in the mesopause and have been reported in the literature for the first time at the end of the 19[th] century (Leslie, 1885). They are optically very thin and can be seen from ground by the naked eye only after the sun has set below the horizon, which is why they are often called noctilucent clouds (NLC). In the recent years NLCs have been intensely studied using ground-based

(e.g. Demissie et al., 2014; Kaifler et al., 2013; Kirkwood et al., 2008) and space-born (e.g. DeLand et al., 2007; Hervig et al., 2016; Rong et al., 2015) methods. Additional studies have shown that during NLC season local temperatures are highly variable with mean temperatures of about 140 K and extremes as low as 100 K (Lübken et al., 2009; Rapp et al., 2002). Typical $H_2O$ concentrations of a few parts per million (Hervig et al., 2009; Seele and Hartogh, 1999) then lead to highly supersaturated conditions, i.e. saturation ratios exceeding $S_h$=100 are frequently observed (Lübken et al., 2009). It is

commonly believed that such high supersaturations in the summer mesopause initiate the heterogeneous formation of



crystalline ice on meteoric smoke particles (MSP) (Gumbel and Megner, 2009; Keesee, 1989; Rapp and Thomas, 2006). MSPs are nano-particles which form by re-condensation of material ablated from meteoroids entering the upper atmosphere (Plane et al., 2015). Recent studies estimated that about 40 t of cosmic material enter the atmosphere each day (Carrillo-Sánchez et al., 2016; Hervig et al., 2017). Approximately 20 % of this material ablates in the upper atmosphere with the

major elemental species being Fe, Mg, and Si (Vondrak et al., 2008). The ablated elemental species then form oxide-, hydroxide-, and carbonate compounds below 85 km and polymerize into nanometer-sized particles (Plane et al., 2015), which are likely present in the form of magnetite ($Fe_3O_4$), wüstite (FeO), magnesiowüstite ($Mg_xFe_{1-x}O$, x=0-0.6), or iron-rich olivine ($Mg_{2x}Fe_{2-2x}SiO_4$, x=0.4-0.5) (Hervig et al., 2017; Rapp et al., 2012). Strong atmospheric circulation limits the average lifetime of these particles in the summer mesopause such that they only reach sizes below about $r$=2 nm (Bardeen et al.,

2008; Megner et al., 2008a; Megner et al., 2008b; Plane et al., 2014). Model simulations have shown, that about 10 % of these particles are negatively charged at NLC height and season (Plane et al., 2015; Plane et al., 2014).

Precise modelling of the formation process of NLCs is, however, hindered by a limited understanding of the microphysical processes involved in heterogeneous ice formation under mesopause conditions. Here, the main unknown parameters are the surface properties (i.e. the ability of the material to serve as ice nuclei) of MSPs, the ice phase forming at mesopause

conditions, and the effect the electrical charge on MSPs may have on the ice formation process. In order to improve our understanding of NLC formation we recently presented an experiment to study ice formation and growth processes on nano-particles exposed to realistic mesopause conditions (Duft et al., 2015). We used this setup in two previous studies to investigate ice growth rates on iron-oxide and silica nano-particles which served as analogues for MSP (Nachbar et al., 2018b, c). We demonstrated that water vapour condenses in the form of amorphous solid water (ASW) at temperatures of the

summer mesopause. In this study we follow up on our recent work and precisely measure onset conditions for the activation of ice growth on small meteoric smoke particle analogues at PMC formation conditions. We performed laboratory experiments by choosing conditions with saturation ratios below and above the activation threshold for ice growth. From these experiments, we determined critical saturations $S_{crit}$ needed to activate ice growth.  We analysed the data considering the formation of ASW and present a new adsorption-activation model, which highly reduces the current uncertainties in

describing ice particle formation in the mesopause.

## 2 Methods

In this work we performed laboratory experiments using the MICE-TRAPS apparatus which was described earlier (Duft et al., 2015; Meinen et al., 2010; Nachbar et al., 2018b; Nachbar et al., 2016). In brief, sub-4nm iron-silicate nano-particles of adjustable elemental composition are produced in a microwave plasma particle source as MSP analogues (Nachbar et al.,

2018a). The nano-particles are transferred continuously to the low pressure ($p<10^{-4}$ mbar) Trapped Reactive Atmospheric Particle Spectrometer (TRAPS) by means of an aerodynamic lens system. Within TRAPS the nano-particles carrying a single positive charge are mass-selected and levitated in the Molecular flow Ice CEll (MICE). MICE is a combination of a





linear quadrupole ion trap and a water vapour supersaturation cell in which pressure, temperature, and humidity conditions of the polar summer mesopause can be established. The nano-particles trapped in MICE are thermalized by collisions with a He background gas. The water vapour partial pressure in MICE is set by temperature controlled sublimation from ice covered surfaces which have been installed in addition to the ion trap electrodes. In this work, the saturation ratio $S$ (short:

saturation) is usually given with respect to the saturation vapour pressure of ASW ($S_{ASW}$). In some cases, and in order to facilitate comparison with previous studies, the saturation ratio is also given with respect to the saturation vapour pressure of hexagonal ice ($S_h$), for which we use the well-established parameterization given by Murphy and Koop (2005). The saturation $S_{ASW}$ can be obtained from $S_h$ using the following relation (Nachbar et al., 2018c):

$$\frac{S_h}{S_{ASW}} = \frac{p_{sat,ASW}}{p_{sat,h}} = \exp\left(\frac{2312\,[Jmol^{-1}] - 1.6\,[Jmol^{-1}K^{-1}]\cdot T}{RT}\right). \tag{1}$$

We use the terms supersaturation and supersaturated conditions for cases in which the saturation is larger than 1.

A typical experiment in MICE starts by filling the particle trap with about $10^7$ size-selected, singly-charged nano-particles in about 1s. Water adsorption and condensation on the trapped nano-particles is monitored by periodically extracting small fractions of the trapped nano-particle population and measuring the nano-particle mass as a function of the trapping time using a time-of-flight mass spectrometer. The saturation is usually varied by changing the particle temperature while keeping

the water vapour density in MICE constant to facilitate comparison. In principle, water vapour density and particle temperature can be chosen independently within the limits of this experimental approach (Duft et al., 2015). Typical mass growth curves of iron oxide particles of initial radius $r_{dry}$=1.87 nm are shown in Fig. 1 for various water vapour saturation ratios between $S_{ASW}$= 1.4 and $S_{ASW}$=16.

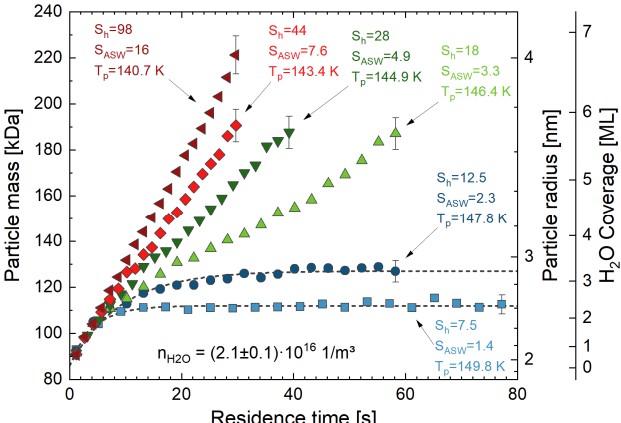

**Figure 1. Water vapour adsorption and depositional growth on Fe₂O₃ nano-particles of initial radius $r_{dry}$=1.87nm under supersaturated conditions. Particle temperature varied between $T$=140.7K at $S_{ASW}$=16 and $T$=149.8K at $S_{ASW}$=1.4. Typical error bars are shown for the last point of each series.**



We note that mass growth occurs independent of the saturation within the first few seconds when using a fixed water vapour concentration. At longer residence times two distinct regimes can be identified:

At low supersaturation ($S_{ASW}$=1.4 and $S_{ASW}$=2.3) the particle mass growth rate decreases with time after the initial growth. The mass accretion due to water adsorption is compensated by an increasing mass loss due to desorption from the particle

surface. The particle mass then approaches a steady equilibrium state after about 25 s for the case shown. This behaviour is observed for all saturation ratios below a certain critical saturation threshold. In the following we will refer to the saturation regime below the critical threshold (i.e. in which the particle mass reaches a steady state) as the equilibrium regime. The mass of adsorbed water molecules $m_{ads}$ in the steady state can be determined by fitting a simple exponential function of the following form to the data

$$m(t) = m_0 + m_{ads}(1 - \exp(-t/\tau)) . \qquad (2)$$

Here, $m$ is the particle mass as function of the residence time $t$, $m_0$ is the initial particle mass, and $\tau$ is the characteristic time for reaching the steady state. Fits of Eq. (2) to the data for $S_{ASW}$=1.4 and $S_{ASW}$=2.3 are shown by the black dashed curves in Fig. 1 resulting in $m_{ads}$=26 kDa And $m_{ads}$=41 kDa, respectively. These values for $m_{ads}$ correspond to 2.3 and 3.2 monolayers of adsorbed water which significantly increases the size of the nano-particles. A parameterization describing the amount of

adsorbed water in equilibrium regime is developed in Sect. 3.1.

At higher supersaturation ($S_{ASW}$=3.3 and above) a continuous particle growth is observed. In this growth regime, condensation always exceeds evaporation from the particle surface. The critical saturation for activation of ice growth $S_{crit}$ is obtained in the experiment by monitoring the conditions at which the transition between equilibrium regime and growth regime occurs (i.e. between $S_{ASW}$=2.3 and $S_{ASW}$=3.3 in Fig. 1). By choosing finer temperature steps the critical saturation can

be determined with higher accuracy. Measured critical saturations will be presented in Sect. 3.2 together with an ice activation model which describes the measured data.

## 3 Results and Discussion

### 3.1 Adsorption in the equilibrium regime

In this section we present a parameterization for the water coverage on iron silicate particles at mesospheric conditions.

Traditionally, the amount of water vapour adsorbed on a surface is described using adsorption isotherms where the water coverage is plotted as function of the saturation ratio (e.g. Venables et al., 1984). The water coverage $\Theta$ is defined as the number of adsorbed water monolayers and is calculated using the wet and dry particle radius as $\Theta=(r_{wet}-r_{dry})/d_{ML}$. For $d_{ML}$ we use the average distance of water molecules in the condensed state $d_{ML}=(m_{H2O}/\rho)^{1/3}$ with molecular mass $m_{H2O}$ and the density of ice ($\rho$=930 kg m$^{-3}$). For hydrophilic materials as iron-oxides and silica it is known that multilayer adsorption

occurs (Mazeina and Navrotsky, 2007; Navrotsky et al., 2008; Sneh et al., 1996).





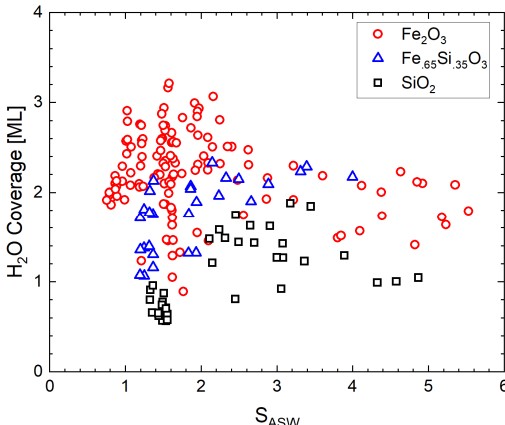

**Figure 2. Water coverage on iron-silicate nano-particles as a function of saturation.**

We measured the mass of adsorbed water vapour in equilibrium regime for iron oxide, silica and mixed iron silicate particles ($r_{dry}$=1.05 – 3.05 nm) covering the temperature range between 128 and 152 K in a total of 192 experiments. We converted

the measured adsorbed mass to $H_2O$ coverages and plot the results in Fig. 2 as a function of $S_{ASW}$. The coverage on iron oxide particles is typically higher compared to iron-silicate and silica particles which is in line with measurements showing that iron oxide exhibits a higher desorption energy than silica (e.g. Mazeina and Navrotsky, 2007; Sneh et al., 1996). We observe, however, that the data does not show the typical multilayer behaviour where coverage increases with saturation. This is a result of the curvature effect, which strongly influences the equilibrium coverage for nanometer-sized particles. Instead, we

find that the adsorption model that we used recently (Nachbar et al., 2018a) to describe the water affinity of iron-silicate particles describes the water coverage more adequately.

The model was originally used to describe the surface concentration of adsorbed monomers at sub-monolayer coverage on a planar surface (Pruppacher and Klett, 2004). It assumes a homogeneous particle surface in which all surface binding sites for adsorbent molecules are characterized by the same surface desorption energy $E^0_{des}$. The equilibrium between desorbing water

molecule flux density $j_{des}$ and adsorbing flux density $j_{ads}$ yields

$$\underbrace{\frac{n \cdot v_{th}}{4}}_{j_{ads}} = \underbrace{c \cdot f \cdot \exp\left(-\frac{E^0_{des}}{RT}\right) \cdot S_K(r_{dry})}_{j_{des}} . \tag{3}$$

Here, $n$ and $v_{th}$ are the number density and the mean thermal velocity of gas phase water molecules, respectively. The number of adsorbed water molecules is described by $c = m_{ads}/(m_{H_2O} \cdot A_{dry})$ with the surface area of the dry particle $A_{dry}=4\pi r_{dry}^2$. The other parameters are the vibrational frequency $f$=10$^{13}$ Hz for $H_2O$ (Pruppacher and Klett, 2004), the universal gas

constant $R$, and particle temperature $T$. We added a Kelvin-effect-like term





$$S_K = \exp\left(\frac{2\sigma M}{RT\rho r_{dry}}\right) \qquad (4)$$

to the flux density of desorbing molecules in order to take the increased desorption due to the curvature effect into account. Here, $M$ is the molar mass of water, $\rho$ is the density of the adsorbed water film which we assume as similar to the density of ice, and $\sigma$ is the interfacial tension between the water film and air. We do not take into account the influence of the particle

charge and of the collision radius of water molecules on the equilibrium saturation. Both effects are small compared to the Kelvin-term and would make the parameterization unnecessarily complicated and would render Eq. (3) analytically not solvable. So we can re-arrange the equation to yield:

$$\underbrace{RT \cdot \ln\left(\frac{4cf}{nv_{th}}\right)}_{\equiv E_{des}} = E_{des}^0 - \frac{2\sigma M}{\rho} \cdot \frac{1}{r_{dry}} \; . \qquad (5)$$

In Fig. 3 we plot the curvature-dependent desorption energy $E_{des}$ as defined by the left-hand-side of Eq. (5) versus the invers

of the dry particle radius using measured data for the adsorbed water vapour mass $m_{ads}$. The figure shows that the three particle materials exhibit different desorption energies and that the water molecules are on average less strongly bound to the particle surface on smaller particles due to the curvature effect.

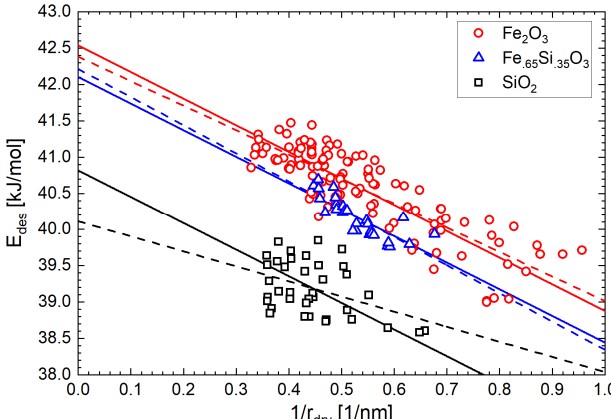

**Figure 3. Surface desorption energy of water molecules on iron silicate nanoparticles calculate using Eq. (5). Dashed lines are**
**independent linear fits and solid lines are linear fits using σ=0.094 Nm$^{-1}$ for all materials.**

In order to determine the surface tension of the water ad-layer we performed independent linear fits to the three data sets which are shown as dashed lines in Fig. 3. The determined surface tensions are $\sigma(H_2O\text{-}Fe_2O_3)=(87\pm5)$ mNm$^{-1}$, $\sigma(H_2O\text{-}Fe_{.65}Si_{.35}O_3)=(99\pm13)$ mNm$^{-1}$, and $\sigma(H_2O\text{-}SiO_2)=(53\pm18)$ mNm$^{-1}$. To keep the intended parameterization for the water coverage as function of the particle material as simple as possible, we refrained from using a material dependent surface

tension of the water ad-layer. We performed a combined fit with a single value for the surface tension for all three materials.





The resulting fits yielded σ=(94±11) mNm⁻¹, which compares very well to the surface tension of ASW at these temperatures (σ$_{ASW}$(T=155 K)=92 mNm⁻¹ and σ$_{ASW}$(T=128 K)=96 mNm⁻¹)(Nachbar et al., 2018c). The fits are shown as solid lines in Fig. 3. From the intercepts of the linear fits we determined the curvature independent desorption energies $E^0_{des}$(Fe₂O₃)=(42.5±0.3) kJmol⁻¹, $E^0_{des}$(Fe.₆₅Si.₃₅O₃)=(42.1±0.2) kJmol⁻¹, and $E^0_{des}$(SiO₂)=(40.8±0.4) kJmol⁻¹. We also re-

5 analysed the adsorption data for mixed iron-silicate particles published in (Nachbar et al., 2018a) using Eq. (5) and σ=94 mNm⁻¹. The resulting curvature independent desorption energies are shown in Fig. 4. Here, the 3 labelled data points are the intercepts from Fig. 3 while all other data points represent single measurements analysed using Eq. (5).

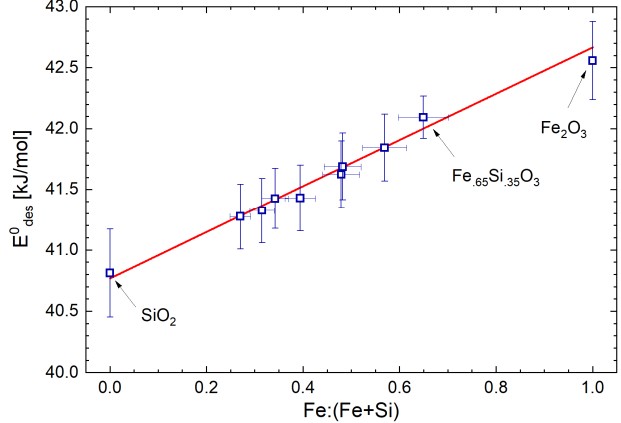

**Figure 4. Curvature-independent average desorption energy as function of iron content of iron silicate particles. The red line**
**represents a linear fit to the data. Based on original data from Nachbar et al. (2018a).**

The average curvature independent desorption energies follow the linear relation:

$$E^0_{des}[\tfrac{kJ}{mol}] = (40.8 \pm 0.1) + (1.899 \pm 0.103) \cdot X , \qquad (6)$$

where X=Fe/(Fe+Si) represents the relative iron content. Inserting this in Eq. (3) we arrive at a parameterization for the water coverage for iron silicate particles at mesopause conditions

$$m_{ads} = A_{dry} m_{H2O} \frac{n \cdot v_{th}}{4f} \cdot \exp\left( \frac{E^0_{des}}{RT} - \frac{2\sigma M}{RT\rho r_{dry}} \right) , \qquad (7)$$

which can be used to predict the mass of adsorbed water for iron-silicate particles as function of the dry particle radius, temperature and water vapour concentration. The wet particle radius $r_{wet}$ and the water coverage Θ can be calculated from $m_{ads}$ using:

$$r^3_{wet} = r^3_{dry} + \frac{3}{4\pi} \frac{m_{ads}}{\rho}; \quad \Theta = \frac{r_{wet} - r_{dry}}{d_{ML}} \qquad (8)$$



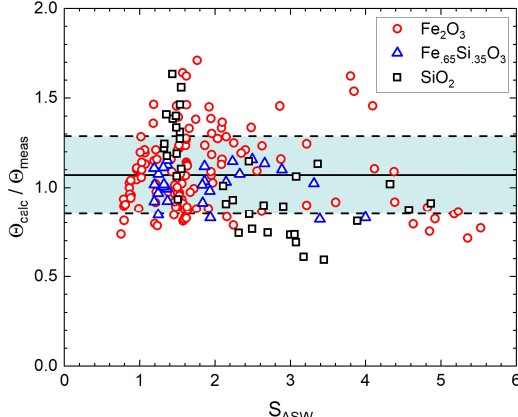

**Figure 5. Ratio of calculated water coverage using Eq. (7) and measured water coverage as function of the saturation ratio.**

In Fig. 5 we show the ratio of calculated to measured water coverages for all three materials. The average calculated water coverage deviates by about 7% from the measurements (solid line) with a mean square deviation of about 22% (dashed lines). The comparison shows that the measured water coverage is well represented by our water adsorption parameterization for particles between $r$=1-3 nm and temperatures between 128 and 152 K.

### 3.2 Critical saturation for ice activation

We have measured the critical saturation for the activation of ice growth on small iron silicate nano-particles ($r_{dry}$=1.05-2.8 nm) in the temperature interval between 128 and 147 K for three different particle compositions in a total of 53 independent experiments. Figure 6a shows the measured critical saturations as a function of the initial dry particle radius for pure iron oxide ($Fe_2O_3$) particles at four different temperatures. Note that we plot here the saturation ratio with respect to the vapour pressure of hexagonal ice to facilitate comparison with other studies. Figure 6b compares the results at 139.6 K with measurements on silica and one mixed iron-silicate with an elemental ratio Fe/(Fe+Si)=0.65. Note that in all measurements ice growth is activated below $S_h$=50, which compares to observations of saturation ratios exceeding $S_h$=100 when temperatures drop below 140 K on a regular basis during NLC season (Lübken et al., 2009). The solid lines in Fig. 6 are the results of an adsorption-activation model which we will present below.



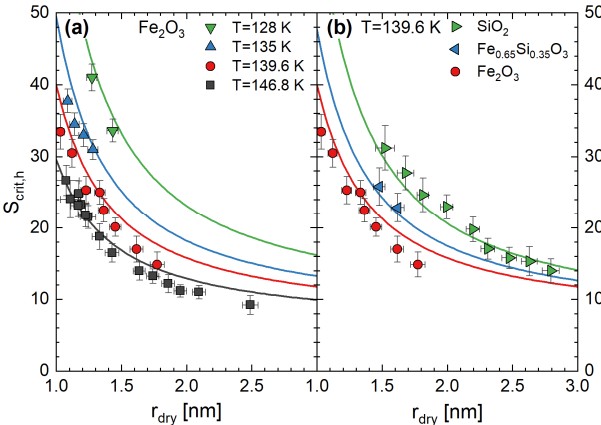

**Figure 6. Critical saturation as function of initial particle radius. Solid lines represent the ice activation model (see text for details).**

We have shown previously that ASW is the initial form of condensed water that deposits on hydrophilic iron silicate nanoparticles at temperatures below 160 K (Nachbar et al., 2018c). Above we showed that these particles are already

covered with more than one monolayer of water at saturations below the ice activation threshold. This water film is well described using bulk properties for the surface tension of ASW. Here, we will rely on a simple approach to describe ice formation in the mesopause by assuming that the nuclei are spherical and perfectly wettable (contact parameter close to one, i.e. the activation barrier to form ASW is low). Under such conditions, and taking into account that the ice particle is charged, ice formation is likely to occur near the equilibrium saturation which is given by the Kelvin-Thomson equation for

the particle radius including the adsorbed water $r_{\mathrm{wet}}$:

$$\ln(S_{KT}) = \frac{M}{RT\rho}\left[\frac{2\sigma}{r_{wet}} - \frac{Q^2}{32\pi^2\varepsilon_0 r_{wet}^4}\left(1 - \frac{1}{\varepsilon_r}\right)\right], \tag{9}$$

where Q is the particle charge, and $\varepsilon_0$ and $\varepsilon_r$ are the permittivity of vacuum and the relative permittivity of water, respectively. For the surface tension of ASW we use $\sigma_{\mathrm{ASW}}=(114.8-0.144\cdot T\,[\mathrm{K}])$ mN m$^{-1}$, a parameterization which is based on a low-temperature extrapolation of measured data for supercooled water (Vinš et al., 2017). Our previous study on the

vapour pressure of ASW indicates that this parameterization is consistent with the properties of ASW to within 10% at the investigated temperatures between 133 and 147 K (Nachbar et al., 2018c). For the density of ASW we use a constant value of $\rho_{\mathrm{ASW}}=0.93$ gcm$^{-3}$ (Brown et al., 1996; Loerting et al., 2011).

In the derivation of the Kelvin-Thomson equation, the vapour in equilibrium with the condensed phase is assumed to be an ideal gas. This also includes the assumption that the gas phase molecules are point-like. This assumption breaks down when

the size of the particle becomes comparable to the size of the water molecules. When taking the size of water molecules into account using a hard sphere collision model the equilibrium saturation changes to:




$$S_{KT}^{*} = S_{KT} \cdot \left(1 + \frac{r_{H2O}}{r_{wet}}\right)^{-2}. \tag{10}$$

The second term on the right-hand-side represents the correction due to the finite size of the water molecules for which we use $r_{H2O}$=1.5 Å (Bickes et al., 1975). Equation (10) reduces to the Kelvin-Thomson term if $r_{H2O}/r$<<1. In our adsorption-activation model the onset conditions for ice growth are reached when the saturation in the environment of the particle

surpasses the equilibrium saturation given by Eq. (5) ($S_{ASW} \geq S_{KT}^{*}$). The adsorption-activation model is illustrated in Fig. 7 for $r_{dry}$=2 nm particles at $T$=140 K. Here, the Kelvin-Thomson radius $r_{wet}$=f$^{-1}$($S_{KT}^{*}$) represents the boundary between equilibrium and ice growth regime (dash-dotted line). For comparison, we also plotted the Kelvin-Thomson radius where we neglected the H$_2$O collision radius (dashed line). Solid lines represent the wet particle radius in equilibrium regime according to Eq. (7) and Eq. (8) for Fe$_2$O$_3$ in red and SiO$_2$ in black. The onset conditions are defined in our model by the saturation ratio $S_{ASW}$ for

which the particle radius in equilibrium regime intersects the Kelvin-Thomson radius.

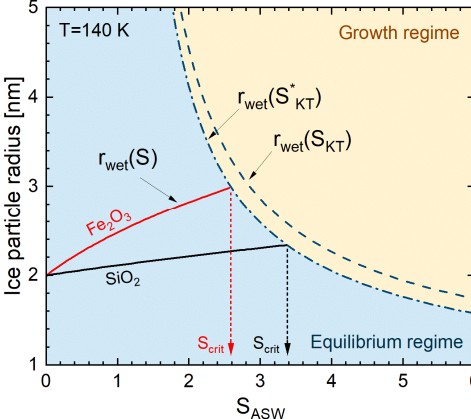

**Figure 7. Illustration of the adsorption-activation model. Solid lines represent the wet particle radius in equilibrium regime calculated using Eq. (7) and Eq. (8). Dash-dotted and dashed lines represent the Kelvin-Thomson radius (Q=1e) calculated with and without H$_2$O collision radius using Eq. (10), respectively.**

We determined the critical saturation ratios using this method and plotted the results as solid lines in Fig. 6 for all temperatures and particle compositions shown. The model curves agree well with measured data confirming the method as a good predictor for ice growth onset at mesopause conditions.



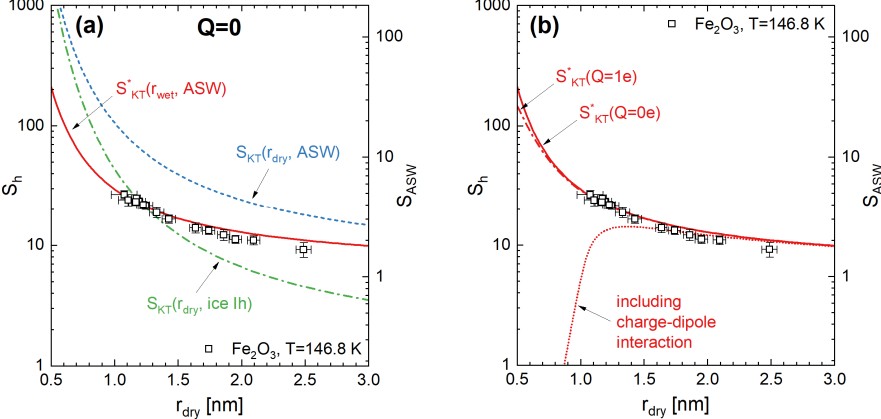

**Figure 8. Comparison of measured onset conditions for ice activation for Fe₂O₃ particles at $T$=146.8 K with different model calculations. Panel a) Comparison of the equilibrium saturation given by the activation model excluding charge effects (Eq. 9, Q=0e). The red solid line represents calculations using the wet particle radius while the blue dashed and green dash-dotted lines represent calculations using the dry particle radius assuming ASW and hexagonal ice, respectively. Panel b) Comparison of the activation model for charged (dash-dotted line, Q=1e) and neutral (solid line, Q=0e) particles. Additionally, the equilibrium saturation including charge-dipole interaction is shown as a dotted line (see text for details).**

In Fig. 8a we plot the results of the adsorption-activation model (solid red curve) for iron oxide particles at 146.8 K. The blue dashed line was calculated using the adsorption activation model but assuming that no water adsorbs prior to activation. The comparison of both model calculations shows that the equilibrium saturation is substantially reduced due to the adsorption of several monolayers of water near the activation threshold. Also shown as a green dash-dotted line is the activation model assuming the formation of hexagonal ice and assuming that no water adsorbs prior to activation. These calculations represent the lowest critical saturations currently assumed in mesospheric models (e.g. Berger and Lübken, 2015; Schmidt et al., 2018). However, it is expected that the majority of MSPs in the mesopause is smaller than $r$=1.2 nm (Bardeen et al., 2008; Megner et al., 2008a; Megner et al., 2008b; Plane et al., 2014). Ice formation on such small particles has to occur in order to explain observed ice particle concentrations in PMCs. According to our activation model, critical saturations are substantially smaller than currently assumed for particles with $r$<1.2 nm.

Figure 8b compares the equilibrium saturation predicted by the activation model for neutral particles (solid curve) with singly charged particles (dash-dotted curve). We note that the presence of several monolayers of water substantially increases the particle radius at the activation threshold which effectively decreases the influence of the particle charge. In consequence, the particle charge becomes significant only below $r_{dry}$=0.6 nm. Note that such small iron oxide particles only consist of about 90 atoms. For comparison, we also plot in Fig. 8b the equilibrium saturation taking into account the additional attraction of polar molecules via charge-dipole interaction according to Nadykto and Yu (2003) which is in disagreement with our measurements below $r_{dry}$=1.5 nm.



### 4 Conclusions

We measured the critical saturation for ice growth on iron-silicate nano-particles serving as meteor smoke substitutes under conditions of PMC formation. Our results show that for iron-silicate particles of dry particle radius $r$=1 nm and above ice growth initiates below $S_h$=50 which is commonly exceeded during NLC season. This affirms meteoric smoke particles as

likely nuclei for heterogeneous ice formation in mesospheric clouds. The onset conditions for ice activation for iron oxide, silica and iron-silicates are well represented by the reduced equilibrium saturation of the wet particle radius using the saturation vapour pressure and surface tension of ASW. This confirms our hypothesis of ASW activation and is in line with our previous observation of ASW depositional growth (Nachbar et al., 2018c). The activation threshold can be matched even more precisely by taking into account the collision radius of water molecules. Our findings show, that that due to the

adsorbed water layer, charge effects play only a minor role in NLC formation for particles larger than a dry size of $r$=0.6 nm. For smaller nano-particles and clusters other competing effects may come into play which could potentially influence the equilibrium saturation, for instance the curvature dependence of the surface tension (Tolman, 1949), or long-range interactions between droplet surface and water molecules (Park et al., 2016; Vasil'ev and Reiss, 1996a, b). Our findings are parameterized in Eqs. (6), (7), and (10) which yield the wet particle diameter and ice cloud activation threshold as a function

of humidity, temperature, dry particle size and iron content.

During the summer season, iron rich particles are heated by absorbed sunlight and it was argued that this would modify their ice activation potential. In an accompanying article (Nachbar et al., 2018d) we show that this is a minor effect under typical mesospheric conditions.

**Author contribution**

DD, MN and TL designed the research, MN and DD performed the measurements and analyzed the data. DD wrote the manuscript with contributions by the co-authors. TL supervised the project.

**Competing interests**

The authors declare that they have no conflict of interest.

**Acknowledgements**

The authors thank the German Federal Ministry of Education and Research (BMBF, grant number 05K13VH3 and 05K16VHB) and the German Research Foundation (DFG, grant number LE 834/4-1) for financial support of this work. We acknowledge support by the Deutsche Forschungsgemeinschaft and Open Access Publishing Fund of Karlsruhe Institute of Technology.




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
