# Peer review of "Unravelling the microphysics of polar mesospheric cloud formation"

_Atmospheric Chemistry and Physics, 2018_

## Referee Comment (RC1) · Anonymous Referee #2 · 5 Dec 2018

This is a very important study. It shows - for the first time - that H2O ice will form on metal silicate particles smaller than 2 nm in radius, at H2O supersaturations that are regularly encountered in the upper mesosphere at high latitudes during summer. The study therefore answers a long-running problem: what is the source of the nuclei for polar mesospheric clouds. The work also shows that the metal oxide/silicate particles do not need to be charged to act as effective ice nuclei under these conditions.

The paper describes a beautiful experimental study, carefully carried out. The precision is such that the authors are able to derive three equations (6, 7 and 10), which determine the ice cloud activation threshold as a function of humidity and temperature, as well as the radius and iron content of the nucleating particle. This is exactly what modellers need to predict where and when these clouds will form, and hence to understand their relevance as a marker for climate change in the middle atmosphere. The authors are to be congratulated on their achievement.

The paper is clearly written and illustrated, and I have only a few grammatical corrections and other minor suggestions which are listed below.

p. 1, line 12: "We observe that ice growth ..." i.e. remove the comma

p. 1, line 13: define NLC

p. 1, line 25: space-borne

p. 2, line 10: "Model simulations have shown that ..." i.e. remove the comma

p. 4, line 13: the method used to estimate the number of monolayers is described in the next section, and should be referred to here e.g. "(see Section 3.1 for method of calculating layer thickness)"

p. 6, line 3: "we assume is similar..."

p. 7, line 5: " in Nachbar et al. (2018a) ..."

p. 11, line 21: perhaps you could suggest here why there is this disagreement?

p. 12, line 9: "Our findings show that due ..." i.e. remove the comma and "that"

---

## Referee Comment (RC2) · Anonymous Referee #1 · 7 Dec 2018

I have just seen the manuscript "The heterogeneous nature of polar mesospheric cloud formation" by Duft, Nachbar and Leisner (DNL). They report water uptake experiments on iron-silicate nanoparticles for various particles sizes and water supersaturation ratios at 128-152 K. This manuscript is part of a series of works by the same three authors - three manuscripts were published in 2016 and 2018, and two manuscripts are reported as submitted. The latter were not available to me at the time of this review. The main conclusion is that noctilucent clouds may form under the experimental conditions in the mesosphere below supersaturations that are typically observed in Earth's upper atmosphere. A secondary conclusion that has already been reported in their earlier manuscripts is that amorphous solid water (ASW) rather than crystalline ice I (cubic, hexagonal or stacking-disordered) condenses on such smoke particles. Finally,

they claim that charge effects or sunlight absorption by the particles is of minor relevance for the formation of mesospheric clouds. An impressive number of almost 200 experiments under different conditions has been performed. However, the data themselves are not convinving, and I have also doubts about the model used to interpret the data as detailed below: 1) My major concern is on the nature of the condensed ASW. The authors treat ASW as if it was a well-defined phase, just like a crystalline phase. However, it is well known that the nature of ASW deposits differs very much depending on the growth conditions. ASW is usually a highly microporous materials. Porosities can be close to 0 or up to 80%, specific surface areas can be between almost 0 and several hundred m2/g. Furthermore, ASW is well known to take up background gases very efficiently and incorporate them into its pores, burying them into its bulk. In fact, some of the micropores in ASW are large enough to take up the whole iron-silicate particle of lenghth scale 1-3 nm in a single pore. Some of this is described by Kay and co-workers (Science 283 1999) as well as by Mayer and Pletzer (J. Phys. Colloques 48 1987 and Nature 319 1986). Unfortunately, any kind of characterization of the ASW particles in terms of porosities, uptake of background gas or additional Fe-Si nanoparticles is missing in this work. Still the authors use specific values in their modelling, e.g., a density of 0.93 g/ml in order to determine the water coverage and wet particle radius. How would the model be affected if the density was much lower than that because of a high porosity within ASW? Even more troublesome is the saturation water vapor of ASW (S_ASW) used in equation (1) of the model. The vapor pressure of ASW maybe much smaller than the one parameterized by Nachbar et al. due to the incorporation of foreign atoms/molecules into the ASW matrix - which automatically implies that the supersaturations given by DNL maybe very different from what they assume in their model. Also the internal porosity and surface corrugation may have a large impact on the vapor pressure. No critical analysis with respect to these uncertainties is discussed in their model. The values they have chosen seem to be taken as granted and are not critically discussed or given the possibility for a range of values for different individual ASW particles - in particular there is no analysis how the results would change if S_ASW and the density were different. 2) The data themselves show a very large scatter. For example in Fig.3 the desorption energies for the SiO2 nanoparticles in the range of 0.3-0.5 nmˆ-1 scatter more or less randomly between 38.5 and 40.0 kJ/mol. With equal justification one could fit a line independent of particle size or that even shows larger desorption energies for smaller particles. The fit given by DNL hinges mainly on the 4 data points between 0.5 and 0.7 nmˆ-1, whereas the other roughly 40 data points do not show any trend. That is, there is a large uncertainty associated with the size-dependent desorption energies deduced from Fig.3 that is not at all reflected in the error-bars in Fig.4. Why is the error-bar for the curvature-independent desorption energy for SiO2 and Fe2O3 in Fig.4 of similar size, even though the data points for Fe2O3 particles scatter much less around the fitted line? A problem that goes hand in hand with this inappropriate fit is the surface-tension deduced from the slope in Fig.3: why is the surface tension for SiO2 half of the one found for Fe2O3 or Fe65Si35O300? One would expect very similar values for all three types of nanoparticles - which demonstrates that the values deduced from Fig.3 are associated with a huge error, much larger than the "+/- 18 mN/M" given by DNL. In my opinion the data for SiO2 do not even allow to exclude a surface tension below 0. Similarly, also in Fig.5 the scatter of the data is unacceptably large. I believe this huge scatter reflects that S_ASW can not be defined in the straightforward way assumed by DNL. In fact, S_ASW needs to be defined individually for each particle, depending on density, curvature and porosity of each deposit. This task is obviously unfeasible, but this still does not validate the use of an invalid assumption. 3) I did not understand from this manuscript why the authors believe that charges do not play a role. Which experimental finding allows for this statement? This needs to be elaborated. Even though the mesosphere contains mostly positively charged ions such as N2+, NO+, O2+,... I would also be interested to see what role negative ions (electrons) might play. Can the authors rule out an influence of electrons on the formation of noctilucent clouds? As the manuscript stands now, I do not feel that it reports important new physics or chemistry that goes beyond what DNL have published in their recent papers on this

topic. The manuscript needs to take into account uncertainties in the model and the nature of the formed ASW with much more care.

---

## Author Comment (AC1) · 6 Feb 2019

**Authors' Response to Reviewers' Comments**

Manuscript No.: acp2018-1018, submitted to ACP
Title: Unravelling the microphysics of polar mesospheric cloud formation
Authors: D. Duft, M. Nachbar, T. Leisner

*The authors would like to thank the reviewers for taking the time to review the manuscript and for preparing detailed and helpful comments.*

**Anonymous Referee #1**

**Comment:**

I have just seen the manuscript "The heterogeneous nature of polar mesospheric cloud formation" by Duft, Nachbar and Leisner (DNL). They report water uptake experiments on iron-silicate nanoparticles for various particles sizes and water supersaturation ratios at 128-152 K. This manuscript is part of a series of works by the same three authors - three manuscripts were published in 2016 and 2018, and two manuscripts are reported as submitted. The latter were not available to me at the time of this review.

**Reply:**

Much to the authors regret, one of the two mentioned manuscripts was accepted and published online one week after initial submission of this manuscript (Nachbar et al., 2018c). It was, however, available after the referee quick reports and before the final manuscript was uploaded for the open discussion. In the uploaded open discussion manuscript the publication is cited with a complete and updated reference. The second submitted manuscript mentioned (Nachbar et al., 2018d) is of minor importance for the results and conclusions of the current manuscript as we only refer to it as an accompanying article in the final paragraph. This manuscript was available as a discussion paper on ACPD before the beginning of the open discussion.

**Comment:**

The main conclusion is that noctilucent clouds may form under the experimental conditions in the mesosphere below supersaturations that are typically observed in Earth's upper atmosphere. A secondary conclusion that has already been reported in their earlier manuscripts is that amorphous solid water (ASW) rather than crystalline ice I (cubic, hexagonal or stacking-disordered) condenses on such smoke particles. Finally, they claim that charge effects or sunlight absorption by the particles is of minor relevance for the formation of mesospheric clouds. An impressive number of almost 200 experiments under different conditions has been performed. However, the data themselves are not convincing, and I have also doubts about the model used to interpret the data as detailed below:

**Remark:**

The following major concerns by the referee do not question the experimental results on the critical saturation. Rather, the critique is directed at the modelling part.

**Comment:**

1) My major concern is on the nature of the condensed ASW. The authors treat ASW as if it was a well-defined phase, just like a crystalline phase. However, it is well known that the nature of ASW deposits differs very much depending on the growth conditions. ASW is usually a highly microporous materials. Porosities can be close to 0 or up to 80%, specific surface areas can be between almost 0 and several hundred m2/g. Furthermore, ASW is well known to take up background gases very

efficiently and incorporate them into its pores, burying them into its bulk. In fact, some of the micropores in ASW are large enough to take up the whole iron-silicate particle of length scale 1-3 nm in a single pore. Some of this is described by Kay and co-workers (Science 283 1999) as well as by Mayer and Pletzer (J. Phys. Colloques 48 1987 and Nature 319 1986). Unfortunately, any kind of characterization of the ASW particles in terms of porosities, uptake of background gas or additional Fe-Si nanoparticles is missing in this work. Still the authors use specific values in their modelling, e.g., a density of 0.93 g/ml in order to determine the water coverage and wet particle radius. How would the model be affected if the density was much lower than that because of a high porosity within ASW?

**Reply:**

The authors agree with the referee that ASW in general is not a well-defined phase as it may be produced in a wide range of experimental growth conditions. It is well known that depending on growth conditions microporous ASW may be formed and that background gas molecules may be trapped if the gas is present during deposition. The most critical parameters for the creation of microporous ASW appear to be deposition angle, deposition rate and substrate temperature (Dohnalek et al., 2003; Hill et al., 2016; Kimmel et al., 2001; Kouchi et al., 1994; Mayer and Pletzer, 1986; Mitterdorfer et al., 2014; Raut et al., 2007; Stevenson et al., 1999). However, highly porous ASW was usually produced using surface temperatures below 77K. ASW samples that were prepared by deposition at surface temperatures between 90 and 110K revealed either only a small degree of porosity (Brown et al., 1996; Chonde et al., 2006), or were nonporous (Kimmel et al., 2001; Stevenson et al., 1999). ASW samples produced above 110K in the absence of a background gas were found to be compact or exhibited the same surface area (determined by gas adsorption experiments) as crystalline ice reference samples indicating the formation of compact ASW (Mayer and Pletzer, 1986; Stevenson et al., 1999; Westley et al., 1998). We thus argue that at the experimental conditions employed in our study (surface temperature of T>=128K, maximum deposition rate 3Å/s (~1ML/s), Helium as a background gas) compact ASW is created with at maximum a marginal degree of porosity. Based on the results of the cited studies we believe that compact ASW can be regarded as a well-defined phase and that it is justified to use specific (temperature-dependent) values for e.g. the density, the vapour pressure, or the surface tension of compact ASW.

**Changes made:**

To address the apparent potential for ambiguity between 'ASW in general' and 'compact ASW', we updated the manuscript and replaced 'ASW' with 'compact ASW' where appropriate. We also added a clarifying statement to section 3.2 (page 9 lines 4-10).

**Comment:**

Even more troublesome is the saturation water vapor of ASW (S_ASW) used in equation (1) of the model. The vapour pressure of ASW maybe much smaller than the one parameterized by Nachbar et al. due to the incorporation of foreign atoms/molecules into the ASW matrix - which automatically implies that the supersaturations given by DNL may be very different from what they assume in their model. Also the internal porosity and surface corrugation may have a large impact on the vapor pressure. No critical analysis with respect to these uncertainties is discussed in their model. The values they have chosen seem to be taken as granted and are not critically discussed or given the possibility for a range of values for different individual ASW particles - in particular there is no analysis how the results would change if S_ASW and the density were different.

**Reply:**

We agree with the referee that the assumption of porous ice with possible incorporation of foreign atoms would render the model inconclusive as large uncertainties in ice density, vapour pressure and surface tension are linked with this assumption. For instance, the surface area as well as the vapour

pressure were found to vary by a factor as large as 10 depending on growth conditions (e.g. Kouchi, 1987; Mayer and Pletzer, 1986; to name just two studies).

However, and as already detailed above, the authors have no indication to assume that the ice formed under the experimental conditions is either porous or has significant amounts of foreign atoms incorporated.

Additionally, the model provides an excellent representation of the experimental data (compare Fig. 6). The standard deviation between model and experimental data including all data points shown is only 9%. The authors would like to emphasize that the model is based on only two main quantities: 1) on a parameterization for the water coverage at mesopause conditions which is based on experimental data shown in this manuscript, and 2) on the Kelvin-Thomson equation in which we use the available specific values for compact ASW. These two quantities alone are sufficient to reproduce measured critical saturations for nanoparticle radii ranging from 1 to 2.8nm, at temperatures between 128 and 147K, $H_2O$ vapour pressures in the range between about $2 \times 10^{-7}$ to $7 \times 10^{-5}$Pa, and covering different iron-silicate compositions. The striking agreement and the simplicity of the model is a compelling support for the model including the underlying assumptions.

**Comment:**

2) The data themselves show a very large scatter. For example in Fig.3 the desorption energies for the SiO2 nanoparticles in the range of 0.3-0.5 nmˆ-1 scatter more or less randomly between 38.5 and 40.0 kJ/mol. With equal justification one could fit a line independent of particle size or that even shows larger desorption energies for smaller particles. The fit given by DNL hinges mainly on the 4 data points between 0.5 and 0.7 nmˆ-1, whereas the other roughly 40 data points do not show any trend. That is, there is a large uncertainty associated with the size-dependent desorption energies deduced from Fig.3 that is not at all reflected in the error-bars in Fig.4. Why is the error-bar for the curvature-independent desorption energy for SiO2 and Fe2O3 in Fig.4 of similar size, even though the data points for Fe2O3 particles scatter much less around the fitted line? A problem that goes hand in hand with this inappropriate fit is the surface-tension deduced from the slope in Fig.3: why is the surface tension for SiO2 half of the one found for Fe2O3 or Fe65Si35O300? One would expect very similar values for all three types of nanoparticles - which demonstrates that the values deduced from Fig.3 are associated with a huge error, much larger than the "+/- 18 mN/M" given by DNL. In my opinion the data for SiO2 do not even allow to exclude a surface tension below 0.

**Reply:**

The authors did not follow an arbitrary procedure when fitting the data shown in Fig. 3 but instead used the basic linear regression function which is implemented in the graphical analysis software OriginPro 2018b by OriginLab Corporation. The employed linear regression algorithm is well documented (URL: https://www.originlab.com/doc/Origin-Help/LR-Algorithm ; last access 23 January 2019) as well as the method for calculating the standard errors of the fit parameters. In the manuscript the authors report the values as retrieved from the analysis software. The fact, that the experimental data for $E_{des}$ of the $SiO_2$ particles shown in Fig. 3 is not strongly correlated with the invers particle radius is directly reflected by the high standard error of the fitted slope for the $SiO_2$ particles. Ultimately, and as stated in the manuscript, a combined fit for all three materials is performed to avoid using a material dependent surface tension for the water ad-layer in Eq. 5. This procedure will result in a single value for the surface tension for the three materials but ignores the possibility that the surface tension of the thin water ad-layer - air interface may be substrate dependent. However, as we show later in the manuscript, this approach is a good simplification for modelling the water coverage. The results in Fig. 5 show that using this procedure we are able to predict the water coverage sufficiently well. Nevertheless, the authors are aware that the model is not fully able to reproduce the variation in measured water coverage. This may hint to possible

deficiencies in the parameterization or to other unknown parameters influencing the water coverage.

The authors agree with the referee that the error bars for the $SiO_2$ data point in Fig. 4 do not reflect the error in the intercept in Fig. 3 when fitting the $SiO_2$ data alone. The error bars in Fig. 4 represent the standard error for the intercept from the combined fit, which implicitly assumes that the slope is identical for all three materials. In the combined fit all data points from the three materials contribute equally with the result that the large quantity of iron oxide data points dominates the final fitted slope and standard error. We believe that the error bars in Fig. 4 are correct under the assumption of a constant surface tension for all three materials.

**Changes made:**
We adapted the caption of Fig. 3 to be more precise.

**Comment:**
Similarly, also in Fig.5 the scatter of the data is unacceptably large. I believe this huge scatter reflects that S_ASW cannot be defined in the straightforward way assumed by DNL. In fact, S_ASW needs to be defined individually for each particle, depending on density, curvature and porosity of each deposit. This task is obviously unfeasible, but this still does not validate the use of an invalid assumption.

**Reply:**
The authors do not agree that the error bars in Fig. 5 are unacceptably large. As stated in the manuscript, calculated and measured water coverages deviate on average by 7% with a mean square deviation of 22%. The authors would like to remind the reader that the presented parameterization covers particle radii ranging from 1 to 3nm, temperatures between 128 and 152K, $H_2O$ vapour pressures between $2x10^{-7}$ and $1x10^{-4}$Pa, and different iron-silicate compositions. The parameterization is intended to be used as a predictor for the water coverage on small iron-silicate nanoparticles in the highly variable environment of the polar mesosphere. In this context the authors believe that this deviation is more than acceptable.

**Comment:**
3) I did not understand from this manuscript why the authors believe that charges do not play a role. Which experimental finding allows for this statement? This needs to be elaborated.

**Reply:**
In the abstract, results and discussion, and conclusion sections the authors state: *"[…] charge effects play only a minor role in NLC formation for particles larger than a dry size of r=0.6nm."*

We agree with the referee that in the manuscript no experimental data for the critical saturation of uncharged particles is presented as proof for the above statement. Instead, only measurements of the activation of ice growth on nanoparticles carrying a single positive charge are presented. However, the authors also present an ice activation model intended to predict critical ice growth conditions. The model is very simple as it is based on the Kelvin-Thomson equation for the wet particle diameter, which is the generalized Kelvin equation for charged liquid particles. Except for the parameterization of the water coverage, no further fitting procedure was employed in the model. As shown in Fig. 6, the experimental data is reproduced excellently by this simple model. The authors interpret this as a strong indication that ice activation under these conditions is governed by the basic energetic principles underlying the Kelvin-Thomson equation.

In the manuscript, the authors expand this interpretation by investigating the influence of the particle charge in the Kelvin-Thomson equation. Comparison of the dash-dotted and solid curves in

Fig. 8b which were obtained by setting Q=1e and Q=0e in their model shows that *in their model* the critical saturation is only marginally influenced by the single charge residing on the particle. It is thus from the model results that we draw this conclusion about "the minor role of charge effects". We agree that in the manuscript the basis for the authors' conclusion on the charge effect is not made sufficiently clear. We therefore modified and adapted the relevant paragraphs as follows:

**Changes made:**
abstract section, page 1, lines 15+
results section, page 11, line 24
conclusion section, page 12, lines 11-13

**Remark:**
While reviewing the corresponding figures and paragraphs concerning the influence of the particle charge we discovered that in Fig. 8b, instead of the model of Yu (2005), the model by Lapshin et al. (2002) was used. We updated Fig. 8 and included the result of our ice activation model using the modified Kelvin-Thomson formulation by Yu (2005) while keeping the result obtained using the model introduced by Lapshin and co-workers.

**Changes made:**
Included an additional model curve in Fig. 8b. Corrected labels, caption and text related to Fig 8b.

**Comment:**
Even though the mesosphere contains mostly positively charged ions such as N2+, NO+, O2+,... I would also be interested to see what role negative ions (electrons) might play. Can the authors rule out an influence of electrons on the formation of noctilucent clouds?

**Reply:**
Regarding the polarity of the particle charge the authors note that in the Kelvin-Thomson equation the particle charge enters as $Q^2$. Hence, in the authors' model the sign of the particle charge does not play a role.

Furthermore, application of the Kelvin-Thomson equation using bulk properties is at least questionable for very small particles in the cluster size regime. The authors therefore cannot rule out an influence of molecular ions or electrons on NLC formation on the basis of the experimental data or the ice activation model as presented in this manuscript.

**Comment:**
As the manuscript stands now, I do not feel that it reports important new physics or chemistry that goes beyond what DNL have published in their recent papers on this topic. The manuscript needs to take into account uncertainties in the model and the nature of the formed ASW with much more care.

**Reply:**
In the manuscript under review, the authors present laboratory experimental data on the critical saturation for ice growth on small nm-sized meteoric smoke analogue particles at conditions similar to those in the polar summer mesopause. The manuscript constitutes the first report on such measurements and the results are direct proof that heterogeneous formation on small nanoparticles is possible at typical mesopause conditions. This result is a strong indication that indeed meteoric smoke particles are the origin for ice particle formation in the mesopause. This in turn strengthens the modelling of NLC formation which heavily depends on the knowledge of the mechanism and the conditions under which ice particle formation may take place in NLCs. The authors believe that the results presented in this manuscript constitute a major advancement in the understanding of ice

formation in the mesopause. The manuscript clearly goes beyond our recent publications, which are 1) on the water affinity and mixing state of iron silicate nanoparticles produced in our setup (Nachbar et al., 2018a), 2) on the vapour pressure of nano-crystalline ice (Nachbar et al., 2018b), and 3) on the vapour pressure of (compact) ASW (Nachbar et al., 2018c).

The authors therefore feel justified to not agree with the referee on this particular comment.

**Anonymous Referee #2**

**Comment:**
This is a very important study. It shows - for the first time - that H2O ice will form on metal silicate particles smaller than 2 nm in radius, at H2O supersaturations that are regularly encountered in the upper mesosphere at high latitudes during summer. The study therefore answers a long-running problem: what is the source of the nuclei for polar mesospheric clouds. The work also shows that the metal oxide/silicate particles do not need to be charged to act as effective ice nuclei under these conditions.

The paper describes a beautiful experimental study, carefully carried out. The precision is such that the authors are able to derive three equations (6, 7 and 10), which determine the ice cloud activation threshold as a function of humidity and temperature, as well as the radius and iron content of the nucleating particle. This is exactly what modellers need to predict where and when these clouds will form, and hence to understand their relevance as a marker for climate change in the middle atmosphere. The authors are to be congratulated on their achievement.

The paper is clearly written and illustrated, and I have only a few grammatical corrections and other minor suggestions which are listed below.

p. 1, line 12: "We observe that ice growth ..." i.e. remove the comma

corrected

p. 1, line 13: define NLC

replaced "NLC" with "polar mesospheric cloud"

p. 1, line 25: space-borne

corrected

p. 2, line 10: "Model simulations have shown that ..." i.e. remove the comma

corrected

p. 4, line 13: the method used to estimate the number of monolayers is described in the next section, and should be referred to here e.g. "(see Section 3.1 for method of calculating layer thickness)"

suggestion adopted

p. 6, line 3: "we assume is similar..."

corrected

p. 7, line 5: " in Nachbar et al. (2018a) ..."

corrected. We also noticed that in this paragraph we incorrectly refer to "the surface tension of ASW" when actually it should read "the surface tension of supercooled liquid water". We changed the statement and the subscripts in the in-line formula accordingly.

p. 11, line 21: perhaps you could suggest here why there is this disagreement?

At the moment, we can only speculate on the reasons for this disagreement. There might be an error involved in the mathematical derivation of the model by Lapshin et al. while we suspect that the ion-dipole interaction is not correctly implemented in both models, by Lapshin et al. and by Yu, respectively. Review of these hypotheses goes beyond the scope of this manuscript and remains as a subject for future investigations.

**Changes made:**
Replaced "equilibrium saturation" with "critical saturation" in the paragraph starting at line 21 on page 11 for consistency with the rest of the manuscript.

p. 12, line 9: "Our findings show that due ..." i.e. remove the comma and "that"

corrected

[revised manuscript text omitted]